# Effects of *Piper betle* L. Extract and Allelochemical Eugenol on Rice and Associated Weeds Germination and Seedling Growth

**DOI:** 10.3390/plants11233384

**Published:** 2022-12-05

**Authors:** Chonnanit Choopayak, Kodchakorn Aranyakanon, Nuttamon Prompakdee, Pranee Nangngam, Anupan Kongbangkerd, Kumrop Ratanasut

**Affiliations:** 1Department of Biochemistry, Faculty of Medical Science, Naresuan University, Phitsanulok 65000, Thailand; 2Center of Excellence in Research for Agricultural Biotechnology, Faculty of Agriculture, Natural Resources and Environment, Naresuan University, Phitsanulok 65000, Thailand; 3Department of Biology, Faculty of Science, Naresuan University, Phitsanulok 65000, Thailand; 4Department of Agricultural Science, Faculty of Agriculture, Natural Resources and Environment, Naresuan University, Phitsanulok 65000, Thailand

**Keywords:** *Piper betle* L., eugenol, allelopathy, paddy weeds, natural herbicide, bioherbicide

## Abstract

Natural herbicide is considered as a sustainable approach for weed management in agriculture. Here, allelopathic activities of *Piper betle* L. extract (BE) and known allelochemical eugenol (EU) were studied against rice and associated weeds in terms of germination and seedling growth. Five plant species including a rice crop (*Oryza sativa* L.); a dicot weed, false daisy (*Eclipta prostrata* (L.) L.); and three monocot weeds, barnyard grass (*Echinochloa crus-galli* (L.) P. Beauv.), swollen fingergrass (*Chloris barbata* Sw.), and weedy rice (*Oryza sativa* f. *spontanea* Roshev.) were studied. The paper-based results demonstrated that BE and EU had inhibitory effects on seed germination and seedling growth. The IC_50_ values of BE and EU for seed germination were ranked from swollen fingergrass, to false daisy, barnyard grass, rice, and weedy rice, respectively. The ratio of root to shoot length of the seedlings indicated that the roots were more affected by the treatments than the shoots. In addition, the gel-based results showed the reduction of the rice seedling root system, especially on lateral root length and the numbers upon the treatments. Taken together, BE had an allelopathic activity similar to that of EU. Interestingly, the major paddy weed, barnyard grass, was more sensitive to BE than rice, underlining BE as a natural herbicide in rice agriculture.

## 1. Introduction

A number of chemicals are commonly used for weed management in agriculture, because of their speed and efficiency. Unfortunately, the synthetic herbicides are poisonous to human health and biodiversity [1,2]. They are directly toxic to farmers and indirectly accumulated in agricultural products [3]. Moreover, they are difficult to eliminate through decomposition, leading to accumulation in soil and water reservoirs, and finally contamination of the food chain system. Approximately 10% of the applied herbicide (about 223 tonnes a.i per year) was modeled to flow into the reservoirs in Nakhon Sawan Province, Thailand [4]. Rice cultivation was the major contributor of herbicide because of the vast area of cultivation. From the report of the office of Agricultural Regulation, Department of Agriculture, Thailand, among the imported pesticides, herbicides are used in the highest quantities and make up the highest value, with a continuously increasing trend [5]. High amounts of herbicide application cause not only high farming costs, but also weed resistance to herbicides worldwide [6,7].

Weeds are the major problem in agricultural performance. They are unwanted plants, competing with crops for nutritional elements from soil. In addition, they are sources of diseases and pesticides, making crops unhealthy and contributing to reduced productivity. Barnyard grass (*Echinochloa crus-galli* (L.) P. Beauv.) and weedy rice (*Oryza sativa* f. *spontanea* Roshev.) are among the most serious weeds in rainwater rice farming [8]. They are widespread in rice fields in the central region to the lower northern region of Thailand. Barnyard grass has become resistant to heavy herbicide application in several ways [9,10,11]. While weedy rice seedlings are indistinguishable from cultivated rice seedlings, they are difficult to be controlled in the rice fields [12].

In order to solve the problem, a new and effective research-evidenced natural herbicide with known mode(s) of action is required. A natural herbicide is a kind of biocontrol for weed management in a cropping system [13,14]. Natural herbicides are considered to be safe for humans and the environment by modulating soil microorganisms, in addition to being readily biodegradable [15]. The herbicidal property is due to allelopathic activity of plants. In nature, allelopathic plants can compete over and grow better than surrounding plants. Several studies about allelopathic activities of plant residues and extracts especially essential oil were reported on weeds [16]. Hemp (*Cannabis sativa* L.) residue chaff lining on soil surface inhibited *Amaranthus tuberculatus* Moq. weed germination [17]. *Eucalyptus globulus* Labill. had phytotoxic effects on germination and early growth of bentgrass (*Agrostis stolonifera* L. cv. Penncross) [18].

Allelopathic activities of plants in nature and plant extracts resulted from allelochemicals contents. There are several groups of allelochemicals including phenolics, terpenoids, flavonoids, and alkaloids [19]. Among them, phenylpropanoids largely exhibited phytotoxic effects. They are phenolic compounds where the molecular structure is composed of a phenol bonding with an allyl group. The functional groups’ substitution at the benzene ring represents the phytotoxicity of the compounds [20]. Isoeugenol, the major constituents in rhizome and root exudate of cogongrass (*Imperata cylindrica* (L.) Beauv.), together with other phenylpropanoids might contribute to this worst invasive alien species [21]. Several studies reported on the phytotoxic effects of eugenol [22,23]. It inhibited *Avena fatua* L. weed germination and strongly suppressed seedling root growth by activating oxidative stress and membrane damage in the root tissue [24]. Furthermore, eugenol-synthesized derivatives caused root system defects together with chromosomal abnormality in the root tips of lettuce (*Lactuca. sativa* L.) seedlings [20].

*Piper betle* L., the common name Betel pepper or Betel vine, is an evergreen perennial climber herb in the Piperaceae family. It is widely grown and has economic importance in South East Asia to South Asia, especially in India [25]. Furthermore, it was considered as a candidate source of a natural herbicide for the following reasons. Firstly, there have been several reports about the allelopathic activity of *P. betle* L. extracts against weed species, with lesser or no effects on crops. The different *P. betle* L. extracts had greater phytotoxic effects on swollen fingergrass (*Chloris barbata* Sw.) and false daisy (*Eclipta prostrata* (L.) L.) weeds than on rice (*Oryza sativa* L.), lettuce (*Lactuca sativa* L.) and Chinese kale (*Brassica oleracea* L.) crops. The betel oil presented the strongest activity on seed germination. At 0.5 mg/mL, betel oil totally inhibited weed germination, but achieved only 20% on crop germination. Meanwhile, ethyl acetate extract presented allelopathic activity similar to that of ethanolic extract. At 1.0 mg/mL, they totally inhibited weed germination, but achieved about 10–20% on crop germination [26]. In addition, Thonsoongnern et al. reported the allelopathic activity of *P. betle* L. extract on mung bean (*Vigna radiata* (L.) R. Wilczek) seed germination via α-amylase activity inhibition [27].

Secondly, *P. betle* L. contains a number of allelochemicals. Woranoot et al. reported that the volatile constituents in the ethyl acetate fraction of dried betel leaf extract, detected by GC-MS, were mainly phenylpropanoids. Among them, allylpyrocatechol-3,4- diacetate was the most abundant, followed by isoeugenol, eugenol acetate and chavicol, respectively [28]. Even though allyl-pyrocatechol was the most abundant compound in the extract, no report mentioned its allelopathic study. In contrast, there are more reports about the allelopathic activity and action of eugenol. Thirdly, *P. betle* L. is a climber, which grows rapidly and produces a large amount of leaf material for the extract. Fourthly, there are few natural enemies, thus avoiding the risk of toxic pesticide application. Lastly, the volatile constituents in *P. betle* L. are microbial-degradable in the environment. Thus, they do not accumulate over long periods in soil and reservoirs.

In spite of the fact that *P. betle* L. demonstrated a potent source of bioherbicide, in order to explore it becoming a successive herbicide, some important points are still missing. Previously, the activities were investigated on seed germination by a paper-based assay in petri dishes and there were only a few instances of serious paddy weeds. Moreover, the mode(s) of *P. betle* L. action is still unknown. In order to gain more insight of *P. betle* L. for natural herbicide development in rice agriculture, in this study, the allelopathic activities of *P. betle*. L. extract (BE) were investigated against rice and paddy weeds, especially on the serious weeds, barnyard grass (*Echinochloa crus-galli* (L.) P. Beauv.) and weedy rice (*Oryza sativa* f. *spontanea* Roshev.). Moreover, early seedling growth and root system development were determined by means of a paper-based and gel-based assay. To disclose the probable mechanism (s) of the BE action, eugenol (EU) was studied together with the BE.

## 2. Results

### 2.1. Weeds Used in This Study

Four weed species used in this study were comprised of three narrow-leaf grasses, including barnyard grass (*Echinochloa crus-galli* (L.) P. Beauv.), swollen fingergrass (*Chloris barbata* Sw.), and weedy rice (*Oryza sativa* f. *spontanea* Roshev.), along with a broad-leaf false daisy weed (*Eclipta prostrata* (L.) L.) (Figure 1).

### 2.2. Phenylpropanoid Constituents in the BE Identified by Gas Chromatography—Mass Spectrometry (GC-MS)

The ethanolic BE of dried betel leaves appeared to be a viscous liquid, dark green, with a pungent odor. The extracted yield was 9.52% (*w*/*w* dried leaves). A GC-MS analysis revealed more than 30 compounds in the BE. According to their chemical structures, the compounds were classified into 5 groups; phenylpropanoids, monoterpenes, sesquiterpenes, alkaloids and benzenoids. Among them, phenylpropanoids were the most abundant (68.94% of total). Since phenylpropanoids were the compounds of interest, we present only the data of phenylpropanoid constituents, which were composed of six compounds. Among phenylpropanoids, allyl-pyrocatechol diacetate (4-allyl-1,2-diacetoxybenzene) was the most abundant (peak 6, 49.68%), followed by eugenol acetate (peak 5, 16.18%). In addition, isomers and derivatives of eugenol could be found, including isoeugenol (peak 3, 1.74%) and methyl eugenol (peak 4, 0.05%), chavicol and its acetate (peaks 1 and 2, 1.29%) (Figure 2 and Appendix A).

### 2.3. Allelopathic Activities of BE on Seed Germination and Seedling Growth by Means of Paper-Plate Assay

The effects of BE on seed germination and seedling growth were monitored by paper-based assay against rice crop and rice-associated weeds. BE had phytotoxic activities towards plant species at different magnitudes in a dose-dependent manner (Figure 3 and Figure 4 and Appendix A). For rice, the IC_50_ of BE for rice germination was 1.41 mg/mL. Under 2 mg/mL BE, the germination percentage of rice was reduced to 14% of the control. According to IC_50_ for germination, swollen fingergrass was the most sensitive weed to BE among the tested samples. The IC_50_ of BE for the swollen fingergrass germination was the lowest at 0.04 mg/mL. Only 8% of the seeds were able to germinate under 0.05 mg/mL BE and there was no germination at all at 0.1 mg/mL BE. The IC_50_ of BE for false daisy germination (0.27 mg/mL) was also lower than that of the rice. Only 11% of the seeds were able to germinate under 0.5 mg/mL BE and there was no germination at 1 mg/mL BE. Interestingly, barnyard grass, which is one of the most problematic weeds in rice fields, was more sensitive to BE than rice. The IC_50_ of BE for barnyard grass germination was 0.85 mg/mL. Only 20% of the seeds were able to germinate under 1 mg/mL and there was no germination at 2 mg/mL BE. However, the IC_50_ of BE for weedy rice was 1.65 mg/mL, which was statistically not different to rice.

In addition to inhibiting seed germination, BE also affected seedling growth. The ratio of root to shoot length (R/S) dropped along with the increasing BE concentration, indicating that the root was the target site of BE action rather than the shoot. At high BE concentrations, the R/S ratio dramatically dropped because of strong radical growth inhibition, causing a lethal effect upon seedlings. For rice, the R/S value dramatically dropped from 63% at 0.5 mg/mL BE to 16% at 1 mg/mL BE treatment. Under 1 mg/mL BE, even though 74% of rice grains were still able to germinate, the seedlings were abnormal, because of undeveloped radicals. For false daisy, under 0.25 mg/mL BE, the R/S value of the seedlings dropped to 28%. Even though 57% of the grains were still able to germinate, the seedlings were abnormal. For barnyard grass, under 0.5 mg/mL BE, the R/S value of the seedlings dropped to 25%. Even though the grains were largely able to germinate (96%), the seedlings were abnormal. For weedy rice, under 1.0 mg/mL BE, the R/S value dropped to 13%. Even though the grains were still largely able to germinate (96%), the seedlings were abnormal (Figure 3 and Appendix A).

### 2.4. Effects of EU on Seed Germination and Seedling Growth by Means of Paper-Plate Assay

The effects of EU on seed germination and seedling growth were quite similar to that of the BE. According to IC_50_ (mg/mL) values of EU for germination, swollen finger grass (0.05) was the most sensitive to EU, followed by barnyard grass (0.23), false daisy (0.24), rice (0.36), and weedy rice (0.46), respectively (Figure 5 and Appendix A).

In addition to seed germination inhibition, EU also inhibited seedling growth, especially on the primary root compared to the shoots. The R/S ratio of the seedlings dropped along with the increasing EU concentration. Under EU at high concentrations, the R/S ratio significantly dropped due to severe radical growth inhibition, causing lethal seedlings. For rice, the R/S ratio sharply dropped from 112% at 0.16 mg/mL to 17% at 0.32 mg/mL. Under 0.32 mg/mL EU, the germinated rice seedlings had stunted radicals, which was lethal. For swollen finger grass, the R/S ratio sharply dropped from 86% at 0.04 mg/mL to zero at 0.08 mg/mL (no germination). For false daisy, the R/S ratio sharply dropped from 79% at 0.04 mg/mL to 26% at 0.08 mg/mL EU. However, the stunted radicals were lethal to seedlings, occurring at 0.16 mg/mL EU. For barnyard grass, the R/S ratio gradually dropped from 111% at 0.08 mg/mL EU to 61% at 0.16 mg/mL and then sharply dropped to zero at 0.32 mg/mL (no germination). For weedy rice, the R/S ratio dropped to 53% at 0.16 mg/mL and then increased to 76% at 0.32 mg/mL because of the shorter shoot length than the root (Figure 5 and Appendix A).

### 2.5. Effects of BE and EU on Rice Seedling Root Systems by Means of In-Gel Assay

Root development of rice sprouts was inhibited by BE and EU in a dose-dependent manner. The concentration of BE that was inhibited by the halving length of the primary root (IC_50-PR_) was 0.053 mg/mL. The length of the primary roots (PR), lateral roots (LR), and crown roots (CR) were shortened, and the numbers and density of LR were decreased under BE treatment (Figure 6 and Appendix A). Similarly, EU inhibited the root system of rice seedlings in a similar manner as BE. The IC_50-PR_ of EU was 0.047 mg/mL. A comparison was made between the BE and EU treatment, even though the phenotypes of root system were quite similar, EU had a little stronger inhibitory effect. The obvious differences between EU and BE treatments were the reduction of CR and LR lengths, while the reduction of PR was not significantly different. Under 0.05 mg/mL treatment, EU inhibited CR length more strongly than BE. In addition, under 0.1 mg/mL treatment, LR density was more inhibited by EU than by BE (Figure 6 and Appendix A).

## 3. Discussion

Natural herbicides are considered as a weed control of choice in cropping, according to allelopathy. Allelopathic plants normally contain several allelochemical compounds, leading to multi-modes of phytotoxic action against weeds. The degree of phytotoxicity is dependent on the amount and kind of allelochemical content. Functional groups and side chains of allelochemicals influence their activities. The methyl group substitution at C-4 of the benzene ring increased the activity, but the hydroxyl group at C2–4 decreased the activity of phenylpropanoid dihydrocinnamic acid against parasitic weed (*Cuscuta campestris* Yunk.) [29]. Among the six coumarin analogs synthesized from eugenol, 8-methoxy-2-oxo-6-(prop-2-en-1-yl)-2H-chromene-3-carboxylic acid presented the highest phytotoxic activity against lettuce seedlings (*L. sativa* L.). It caused stunted roots and chromosomal abnormalities in the root tips of the seedlings [20].

Because the extraction methods resulted in different kinds and amounts of chemical constituents in the extracts, they influenced the allelopathic activity of the extracts as well. In a comparison between extraction methods, hydrodistillation from fresh leaves and ethanolic extraction from dried leaves, the P. betle L. extracts contained quite similar kinds of compounds, mainly phenylpropanoids, but in different amounts. Whereas the ethanolic extract contained phenylpropanoids at 79.48%, the betel oil (from hydrodistillation) contained at 66.1%. While the eugenol and derivatives were found equally between the methods (about 50%), allyl-pyrocatechol was found to be higher in ethanolic extract (30.9%) than in betel oil (14.4%). Moreover, the extract yield by hydrodistillation was very low (0.25% (*w*/*w* of fresh leaves), compared to that by ethanolic extraction (9% (*w*/*w* of dried leaves) [26]. In this study, the *P. betle* L. ethanolic extract (BE) was prepared by macerating dried betel leaves in absolute ethanol. After the ethanol was completely removed using a rotary evaporator, volatile constituents were identified by GC-MS, separating in a 5% phenyl—methylpolysiloxane nonpolar capillary column. Our findings demonstrated that the major volatiles in the BE were phenylpropanoids (68.94% of total). Among them, allyl-pyrocatechol diacetate was the most abundant (49.68%), followed by eugenol and derivatives (18%) and chavicol and derivatives (1.29%), respectively. Compared to a previous study, the major compounds in *P. betle* extracts were the same as the phenylpropanoids eugenol and allyl-pyrocatechol. However, the amounts of compounds were differential, and eugenol and derivatives were lower in this study. This might be the result of several factors: the stages of leaf materials, the season of collecting, the process of leaves drying, the process of extraction, etc..

According to the chemical structure, the simplest structure of the phenylpropanoid is chavicol. It is composed of phenol bonding with an allyl group. The chavicol substitution by one more hydroxyl group becomes allyl-pyrocatechol. The substitution by one more methoxyl group becomes eugenol or isoeugenol. However, it is not clear how the allelopathic activity of chavicol changes among these isoforms. For the allelopathic study in this work, eugenol was chosen as the comparative allelochemical compound to the BE because of the studies of allelopathic activity and the mode of action of eugenol or isoeugenol in several reports [20,24,30]. Nevertheless, the most abundant phenylpropanoid in BE was allyl-pyrocatechol diacetate (4-allyl-1,2-diacetoxybenzene), and the allelopathic action of BE might be from this compound. Unfortunately, no evidence about the allelopathic study of allyl-pyrocatechol has been reported to this point.

Here, the allelopathic activity of the extract on seed germination and seedling growth on paper in petri dishes was investigated. Our results demonstrated the differential allelopathic activities of the betel extract (BE) among plant species. The inhibitory effect increased in a dose-dependent manner with the different concentrations that halved the seed germination (IC_50_) among the tested plants compared to the solvent control group. The IC_50_ (mg/mL) ranked from the least (the most vulnerable to BE) to the most (the most tolerant to BE) as: swollen fingergrass < false daisy < barnyard grass < rice < weedy rice, indicating that swollen fingergrass, false daisy, and barnyard grass weeds were more sensitive to BE than rice. These results are consistent with previous studies showing that *P. betle* L. extracted by ethyl acetate inhibited the germination of swollen fingergrass seeds and false daisy seeds without affecting the germination of crops [26].

Interestingly, BE at 0.5 mg/mL was much more phytotoxic to swollen fingergrass, false daisy, and the serious paddy weed, barnyard grass, than to rice. There was no effect on rice seed germination, and the rice seedlings were normal. BE totally inhibited swollen fingergrass seed germination and the 11% germinated false daisy showed lethality to seedlings. Even though the barnyard grass seeds could germinate, the seedlings were lethal due to unelongated radicals. Since barnyard grass is the most problematic weed in paddy fields, as a widespread and herbicide-resistant plant with multiple modes^9^, the allelopathic activity of BE over barnyard grass germination and seedling growth emphasized the potential of BE as a natural herbicide in rice fields.

To clarify the target of inhibition, the growth and development of rice seedling roots was evaluated under BE treatment by means of an in-gel assay. Our results demonstrated that the root system of the rice seedlings was inhibited under BE in a dose-dependent manner. The root system did not develop properly, especially in terms of the lengths, numbers, and patterns of LR and CR. This was similar for the EU-treated seedlings. Several reports mentioned the root system as the target of phenylpropanoid allelochemicals [31]. For monocot species, the fibrous root system is vital for plant growth. The big and strong root system is responsible for competition, stress tolerance and productivity. Thus, allelochemicals are prone to disrupt the proper root system growth and development, especially at the early stage of seedlings, leading to undeveloped seedling and, finally, death.

The mechanisms of BE action were due to the allelochemicals. The major volatile constituents in the BE characterized by GC-MS were phenylpropanoids. Among them, allyl-pyrocatechol diacetate was found in the most abundant percentage. Thus, the compound might represent the phytotoxic activity of BE. Nonetheless, no evidence about its allelopathic effect was reported. In this work, the allelochemical eugenol was chosen for comparative study together with BE in order to characterize the allelopathic action of BE. In this case, eugenol was chosen because it was found in substantial amounts in the BE. Moreover, eugenol was previously reported as an allelochemical with known modes of action. Eugenol-rich extracts of *Scrophularia striata* Boiss. ecotype Lizan could inhibit seed germination and reduce the root length of the invasive weed *Chenopodium album* L. [32]. Our results suggested that the allelopathic action of BE results mainly from eugenol, because the phenotypes of BE-treated plants and EU-treated plants were similar with different magnitudes. Although the mechanisms of BE action have yet to be studied, these might be proposed according to the allelochemical eugenol. However, allyl-pyrocatechol and other unreported compounds including terpenoid and alkaloid volatiles were not excluded for the molecular action of BE. Ahuja et al. reported that eugenol inhibited root development through the tissue damaging mechanism by ROS [24]. The mechanism of eugenol action involved ROS-mediated oxidative stress to the roots. Moreover, eugenol inhibited green algae *Ulva prolifera* O. F. Muller aquatic weed growth by introducing oxidative damage and the photosynthetic electron transport rate of photosystem II (rETR) reduction [22]. In addition to proposing the mechanism of action based on allelochemical contents, these actions might be related to phenotypes. Cheng et al. reported the inhibitory activity of Garlic Diallyl disulfide on the seed germination and root length of tomato by regulating cell division, disturbing auxin balance, and expanding gene expression at the root tip, resulting in root cell proliferation and elongation inhibition [33]. Dihydrocoumarin caused root cell membrane disruption and abnormal seedlings by disrupting plant hormones and phenylpropanoid biosynthesis in barnyard grass [31]. A transcriptomics analysis revealed the relationship between phenylpropanoids biosynthesis and root growth development in rice by quercetin treatment [34].

According to our findings, BE demonstrated allelopathic activity on weeds at early stages of growth, suggesting that it might be a potent weed control agent in the rice fields. However, to develop BE until it can actually be successfully applied in the field, more issues should be addressed [13]. For instance, the co-planting experiment between rice and weeds should be considered under the BE application in the field in order to investigate the effect of the extract. In addition, the interaction between BE, soil particles and available nutrients to be absorbed into crops under BE treatment should be studied. The mechanisms of BE action(s) and plant response to an individual BE and the combination with synthetic chemicals need to be further evaluated prior to subsequent application.

## 4. Materials and Methods

### 4.1. Weed Identification, Seeds Collection and Breaking Dormancy

The rice (*Oryza sativa* L.) variety Phitsanulok 2 seeds were provided by the Rice Research Center, Wang Thong District, Phitsanulok province, Thailand. The weeds were collected from paddy fields around Naresuan University, in Phitsanulok province. The specimens were identified by a plant taxonomist. The voucher weed specimens were registered and deposited in a plant specimen collecting room at Naresuan University, Phitsanulok province, Thailand. Meanwhile, the mature seeds were collected, air-dried, and then kept in a paper bag under room temperature.

Prior to allelopathic assay, the seeds in which the germination rate in distilled water was less than 80% were subjected to breaking dormancy in order to increase the germinate rate to over 80%. For the swollen fingergrass and false daisy dormancy breaking process, the seeds were incubated at 50 °C in a hot air oven for 12 h. For the barnyard grass dormancy breaking process, the combination treatments of sulfuric acid followed by potassium nitrate were performed. The seeds were soaked in 98% sulfuric acid for 15 min. After washing in sterilized distilled water three times, the sink seeds in distilled water were collected and further incubated in 50 mM KNO_3_ for 16 h. and then re-washed in distilled water.

### 4.2. Eugenol (EU) Preparation

Eugenol was commercially purchased from the Acros organics company (New Jersey, USA). Since it was slightly dissolved in polar solvent, the stock eugenol was dissolved firstly in absolute ethanol (AR grade, MERCK), and afterwards in Tween-80. The starter solution for assays was 0.5 M eugenol in 37.5% tween-80 and 16.67% ethanol.

### 4.3. P. betle *L.* (BE) Extraction

The *P. betle* L. used in this study was cultivated in Naresuan Univeristy, Phitsanulok province, Thailand. The medium-sized, healthy leaves were collected, thoroughly washed in tap water, and dried in a hot air oven at 50 °C. The small pieces of dried leaves were fermented in 99% ethanol at a ratio of 1:10 (*w*/*v*) in a closed system glass jar and kept in the dark at room temperature for 2 days. The filtrate (through Whatman filter paper No.1) was then collected. The solvent was completely removed using a vacuum Heidolph^TM^ Hei-VAP value rotary evaporator (Heidolph Instruments GmbH & Co. KG, Schwabach, Germany). The crude extract was weighed and kept in a dark brown closed bottle at 4 °C.

### 4.4. Determination of Phenylpropanoid Constituents in the BE by GC-MS Technique

The BE was dissolved in 99% ethanol before subjecting it to an Agilent Technologies GC-MS system, a GC Agilent 6890N and an MS Agilent 2577A 5973N using an HP-5MS capillary column (30 m × 0.25 mm ID, film thickness 0.25 µm). Under conditions in which the cabinet was set to 50 °C for 0.2 min, the temperature was then increased at a rate of 10 °C min^−1^ to 200 °C, converted to an increment of 30 °C min^−1^ to 245 °C, 280 °C for 5 min, and the front inlet temperature was set to 100 °C. The carrier gas was helium with a flow rate of 1.0 mL min^−1^. Mass data was collected in the range of 50–300 m/z. The phenylpropanoid compounds were identified by comparing the mass spectra of the peak to the Wiley Mass Spectral library database. The compound quantification was calculated according to peak area and reported as percentages of the total detected volatiles.

### 4.5. Seed Germination and Seedling Growth Assay by Means of the Paper-Plate Method

The allelopathic effects of BE and EU on seed germination and seedling growth of rice and four weed species were evaluated by the paper-based method. According to the different size of the seeds, about 20–30 seeds were placed on the sterilized germination paper type K-1 (Kian Gwan Public Company Limited, Thailand) in individual glass petri dishes (9 cm diameter) containing 5 mL BE or EU at various concentrations using the diluent of the treatments as controls (BE treatment control: 0.5% tween-80; EU treatment control: the combination of ethanol and tween-80 at different concentrations). The plates were sealed with parafilm and then placed in a plant growth room, under a growth condition of 25 ± 2 °C, 65% relative humidity, light intensity 20 μmol (m^2^)^−1^ s^−1^, and 12/12 light periodic cycle. More than three replications were performed for each treatment. The number of germinated seeds were counted daily for 7 days. The germinated ones were identified based on the radical protruding more than 2 mm. The germination data was calculated and reported as the percentage of germination and concentration that inhibited the half of germination (IC_50_) compared to the control. The root length and shoot height of the seedlings were measured using ImageJ software ver. 15.3t (Wisconsin, USA). The seedling growth data was reported as the relative percentage of ratio between root to shoot length (R/S) of the treatment to the control.

### 4.6. Rice Seedling Root System Determination by Means of In-Gel Method

The inhibitory effects of BE and EU on the rice seedling root system were evaluated by a gel-based method. The rice seeds without husk were sterilized using 2.5% sodium hypochlorite for 15 min. After washing, the seeds were allowed to germinate in sterile distilled water for 18 h. Afterwards, a seed was placed on the surface of ½ MS with 0.25% gellan gum (CultureGel™, USA) gel medium containing BE, EU or solvent control in a glass test tube size 150 × 16 mm (a seed per a tube). The tubes were tightly closed up with aluminum foil and placed in a plant tissue culture room under a growth condition of 25 ± 2 °C, relative humidity at 65%, light intensity 20 μmol (m^2^)^−1^ s^−1^, and light periodic cycle 12/12 until seedlings developed a complete root system (about 7 days). Ten seeds were tested in each replication. More than three replications were performed for each treatment. The length and number of roots were counted and measured daily using ImageJ software ver 1.53 t. The obtained values were statistically calculated and reported as the concentrations of BE or EU capable of halving the length of PR under treatments, comparing to the control (IC50-PR).

### 4.7. Statistical Analysis

A statistical analysis was performed using the Statistical Product and Service Solution software (SPSS Inc., Chicago, IL, USA) version 23.0 and One-Way Analysis of Variance (ANOVA). The results were expressed as the mean ± standard deviation (SD). Duncan’s Multiple Range Test was used to determine significant differences between the mean at *p* < 0.05, *p* < 0.01.

## 5. Conclusions

The *P. betle* L. extract had greater allelopathic activity on paddy weeds than on rice. It inhibited the germination and development of weed sprouts on paper plate and in-gel based assays. Interestingly, 0.5 mg/mL *P. betle* L. had little effect on rice seedling growth, but it was able to control the barnyard grass that is a major problem in rice farming. The inhibitory effect of BE was on root development, especially lateral roots and crown roots similar to eugenol. This shows that *P. betle* L. is a potential natural herbicide that is friendly to humans and the environment.

## Figures and Tables

**Figure 1 plants-11-03384-f001:**
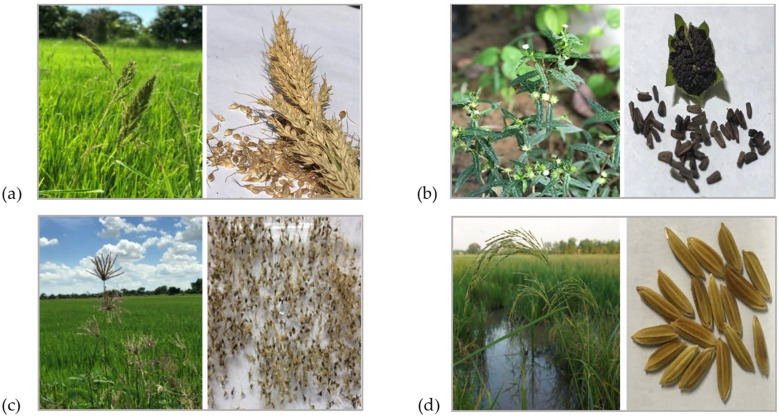
The characteristics of paddy weeds and seeds used in this study. (**a**) barnyard grass (*Echinochloa crus-galli* (L.) P. Beauv.) (**b**) false daisy (*Eclipta prostrata* (L.) L.) (**c**) swollen fingergrass (*Chloris barbata* Sw.) (**d**) weedy rice (*Oryza sativa* f. *spontanea* Roshev.).

**Figure 2 plants-11-03384-f002:**
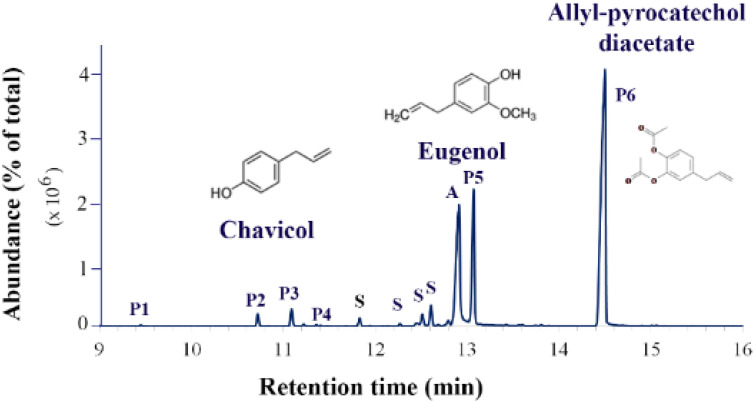
Phenylpropanoid constituents detected in the BE using GC-MS. Peak P1, chavicol; peak P2, chavicol acetate; peak P3, isoeugenol; peak P4, methyl eugenol; peak P5, eugenol acetate; peak P6, allyl-pyrocatechol diacetate. S, sesquiterpenes; A, alkaloid.

**Figure 3 plants-11-03384-f003:**
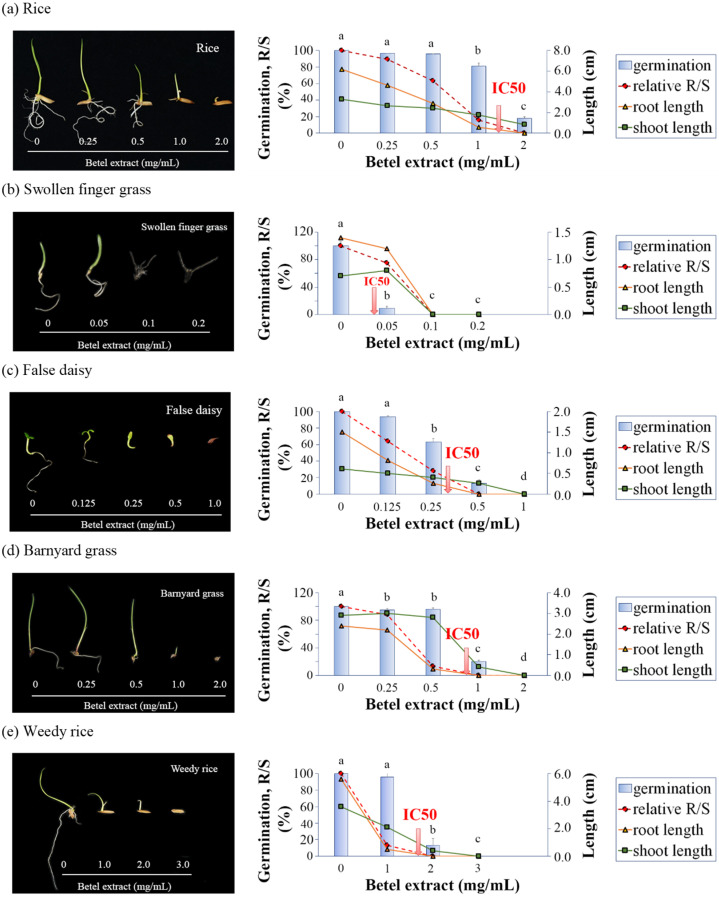
Effects of BE on seed germination and seven days after germination seedling growth by means of paper-plate assay. (**a**) rice (*O. sativa* L.) (**b**) swollen fingergrass (*C. barbata* Sw.) (**c**) false daisy (*E. prostrata* (L.) L.) (**d**) barnyard grass (*E. crus-galli* (L.) P. Beauv.) (**e**) weedy rice (*O. sativa* f. *spontanea* Roshev.). The data in the graphs represented the photo of seedlings in the left panel. The arrows in the graphs indicated the IC_50_ values for germination inhibition. Values are means ± SD of three replications (*n* > 100). Different superscript letters on the bars indicated statistically significant differences of germination among different BE concentrations at *p* < 0.05.

**Figure 4 plants-11-03384-f004:**
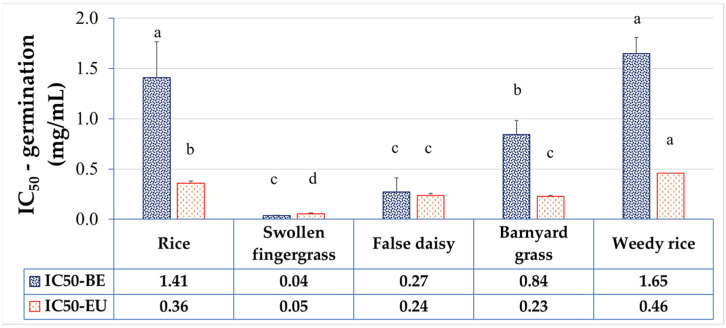
IC_50_ values of BE and EU on seed germination by means of paper-plate assay. Values are means ± SD of three replications (*n* > 100). Different superscript letters on the bars indicated statistically significant differences of IC_50_ among plant species at *p* < 0.05.

**Figure 5 plants-11-03384-f005:**
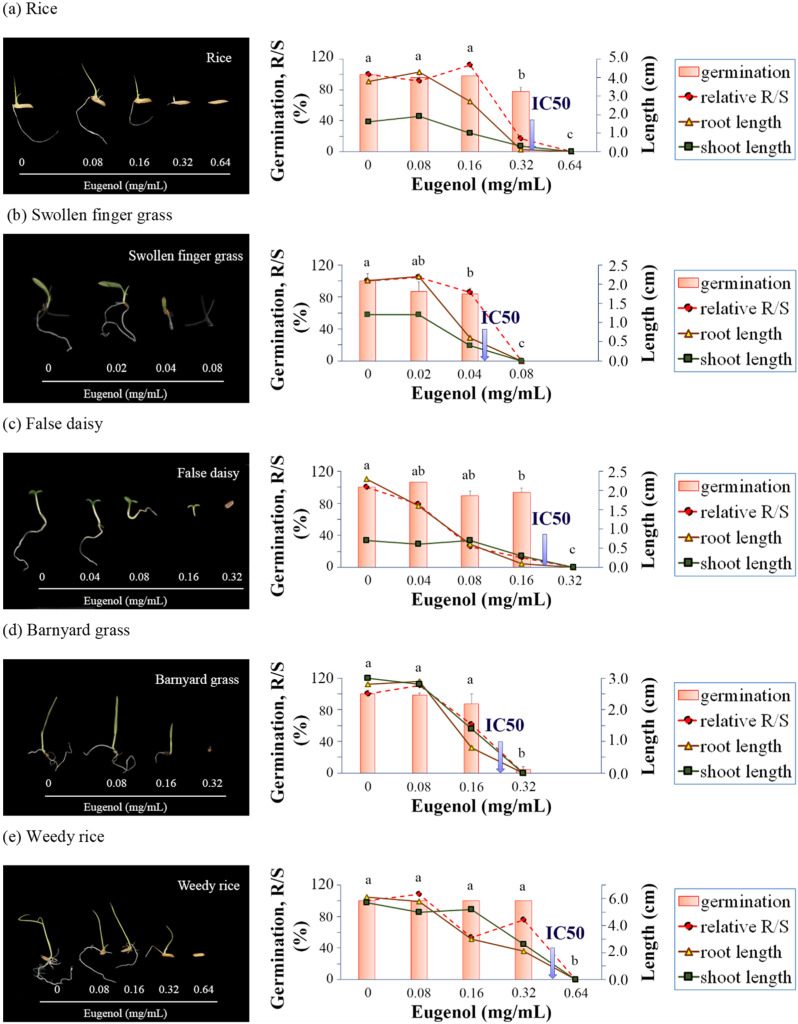
Effects of EU on seed germination and 7 days after germination by means of paper-plate assay. (**a**) rice (*O. sativa* L.) (**b**) swollen fingergrass (*C. barbata* Sw.) (**c**) false daisy (*E. prostrata* (L.) L.) (**d**) barnyard grass (*E. crus-galli* (L.) P. Beauv.) (**e**) weedy rice (*O. sativa* f. *spontanea* Roshev.). The data in the graphs represent the photo of seedlings in the left panel. The arrows in the graphs indicate the IC_50_ values for germination inhibition. Values are means ± SD of three replications (*n* > 100). Different superscript letters on the bars indicated statistically significant differences of germination among different EU concentrations at *p* < 0.05.

**Figure 6 plants-11-03384-f006:**
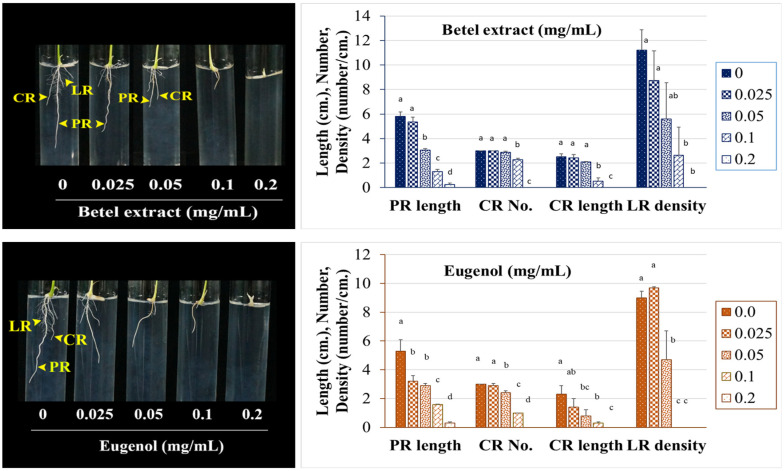
Effects of BE and EU on rice seedling root growth and development by means of in-gel assay. Values are means ± SD of three replications (*n* > 30). Different superscript letters on the bars indicated statistically significant differences of values among different concentrations at *p* < 0.05. PR: primary root; LR: lateral root; CR: crown root.

## Data Availability

Not applicable.

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
