# Peer review of "Effects of Piper betle L. Extract and Allelochemical Eugenol on Rice and Associated Weeds Germination and Seedling Growth"

_plants, 2022, doi:10.3390/plants11233384_

Round 1
Reviewer 1 Report
1- Section 2.2 need extensive correction, the text doesn't speak to the figure.
2- the method used for material preparation it has a major mistake, where the authors heated/dried the plant at 50 oC for two days, this condition is enough to destroy/remove the volatile constituents, which indicate a serious problem in oil produced and if it represent the natural one, the authors may prepare the oil before washing and drying and compare the gc-ms data and if it matching.
3- the chemical constituents is poorly discussed and reference 9 is not the perfect one to be selected, the chemistry section need to be discussed in details and comparing with other related studies.
Author Response
Dear Review 1,
Thank you very much for your comment and suggestion, this is very useful for me. let me considering in these points and precise explanation and discussion.
Please see the attachment
Sincerely yours,
Chonnanit Choopayak

Reviewer 2 Report
Dear authors, this paper is interesting and also this study represent an hot topic to improve sustainability in agriculture. The outset of the manuscript is good, however, some changes are required in order to improver the manuscript readibility and publish it.
- Abstract
few things need to be checked (see the file attached)
- Introduction
introduction should be implemented with additional informations about weeds studied and their noxious effects on rice. Moreover, a further effort should be done to provide evidence of the advantages of using natural-based herbicides.
- Results
results are clearly presented, only few things to check
- Discussion
In this section authors should provide a sufficiently thorough discussion in which they compare findings to current literature in order to prove the points of their study. Please delete some unecessary information and provide more literature to compare these findings.
from line 168 to line 183 authors should consider to move this section in the introduction or in materials and methods since thery are explainig why BE was chosen n for this experiment.
from line 184 to 198 authors are explaining the germination process of monocotyledonous plants. Please delete all the unecessary information
- Materials and methods
authors should make and effort to clearly describe the experimental design of the seed germination and seedling experiments.
- Conclusion
conclusion is well written
Additional details are reported in the file attached.

Author Response
Thank you very much for your comment, suggestion and your elaborate correction for my manuscript. Thank you for the excellent opportunity, providing me to learn something very useful. let me considering in these points and precise explanation and discussion. I corrected all the points as your suggestions.
Please see the attachment
Sincerely yours:
Chonnanit Choopayak

Reviewer 3 Report
In the present work, allelopathic activities of Piper betle extract (BE) and known allelochemical eugenol (EU) were studied against rice and associated weeds in terms of germination and seedling growth. Five plant species including a rice, a dicot weed, and 3 monocot weeds were studied. The paper-based results demonstrated that BE and EU had inhibitory effects on seed germination and seedling growth. The IC50 values of BE and EU for seed germination were ranked from swollen finger grass, to false daisy, barnyard grass, rice, and weedy rice respectively. The ratio of root to shoot length of the seedlings indicated that the roots were more affected by the treatments than the shoots. In addition, the gel-based results showed the reduction of rice seedling root development, especially on lateral root length upon the treatments. Taken together, BE had allelopathic activity similar to that of EU. Interestingly, the major paddy weed, barnyard grass, was more sensitive to BE than rice, underlining BE 23 as a natural herbicide in rice agriculture. In general, this work was well conducted and exhibits interesting results. However, in my opinion, some minor points must be revised after final acceptance, as follow:
1. Some details concerning the phytochemical point of view and association with the allelopathic activity of Piper betle must be included in the Introduction.
2. It´s not clear to me why the authors decide to use GC/MS in the chemical dereplication of a crude EtOH extract. The use of HPLC/MS will be more adequate.
3. Figure 2 – there are several non-identified peaks between 4 and 5 (this last is one of the more abundant compounds!). Why these compounds were not adequately characterized? This point must be carefully revised.
4. Several studies were performed using eugenol – as reported in figures 5 and 6, for example. Why eugenol was chosen for the conduction of these assays? Allylpyrocatechol diacetate was one of the main identified phenylpropanoids in crude extract and was not selected for the conduction of similar assays.
Author Response
Dear Reviewer 3,
Thank you very much for your comments and suggestion. This is very useful for me, let me considering in these points and precise explanation and discussion.
Please see the attachment
Sincerely yours,
Chonnanit Choopayak

Round 2
Reviewer 1 Report
The article is improved, thanks
Reviewer 2 Report
Dear authors,
congratulations for the great work. All the comments have been correctly addressed and the manuscript is thoroughly improved.